# The Distribution and Composition of Vector Abundance in Hanoi City, Vietnam: Association with Livestock Keeping and Flavivirus Detection

**DOI:** 10.3390/v13112291

**Published:** 2021-11-16

**Authors:** Thang Nguyen-Tien, Anh Ngoc Bui, Jiaxin Ling, Son Tran-Hai, Long Pham-Thanh, Vuong Nghia Bui, Tung Duy Dao, Thuy Thi Hoang, Lieu Thi Vu, Phong Vu Tran, Duoc Trong Vu, Åke Lundkvist, Hung Nguyen-Viet, Ulf Magnusson, Johanna Frida Lindahl

**Affiliations:** 1Zoonosis Science Center, Department of Medical Biochemistry and Microbiology, Uppsala University, 75237 Uppsala, Sweden; jiaxin.ling@imbim.uu.se (J.L.); ptlong.vndah@gmail.com (L.P.-T.); ake.lundkvist@imbim.uu.se (Å.L.); J.Lindahl@cgiar.org (J.F.L.); 2International Livestock Research Institute (ILRI), Hanoi 10000, Vietnam; H.Nguyen@cgiar.org; 3National Institute of Veterinary Research, Hanoi 10000, Vietnam; buingocanh_1980@yahoo.com (A.N.B.); buinghiavuong@gmail.com (V.N.B.); ddtung83@yahoo.com (T.D.D.); hongthuy9519@gmail.com (T.T.H.); 4National Institute of Hygiene and Epidemiology, Hanoi 10000, Vietnam; haison20284@yahoo.com (S.T.-H.); coxanh.v@gmail.com (L.T.V.); tvp@nihe.org.vn (P.V.T.); vu.duoc@gmail.com (D.T.V.); 5Department of Animal Health, Ministry of Agriculture and Rural Development, Hanoi 10000, Vietnam; 6Department of Clinical Sciences, Swedish University of Agricultural Sciences, 75007 Uppsala, Sweden; Ulf.Magnusson@slu.se

**Keywords:** vector abundance, urban livestock keeping, mosquito-borne flavivirus, Japanese Encephalitis virus, Hanoi city

## Abstract

Background: Dengue virus and Japanese encephalitis virus are two common flaviviruses that are spread widely by *Aedes* and *Culex* mosquitoes. Livestock keeping is vital for cities; however, it can pose the risk of increasing the mosquito population. Our study explored how livestock keeping in and around a large city is associated with the presence of mosquitoes and the risk of them spreading flaviviruses. Methods: An entomological study was conducted in 6 districts with 233 households with livestock, and 280 households without livestock, in Hanoi city. BG-Sentinel traps and CDC light traps were used to collect mosquitoes close to animal farms and human habitats. Adult mosquitoes were counted, identified to species level, and grouped into 385 pools, which were screened for flaviviruses using a pan-flavivirus qPCR protocol and sequencing. Results: A total of 12,861 adult mosquitoes were collected at the 513 households, with 5 different genera collected, of which the *Culex* genus was the most abundant. Our study found that there was a positive association between livestock keeping and the size of the mosquito population—most predominantly between pig rearing and *Culex* species (*p* < 0.001). One pool of *Cx. tritaeniorhynchus*, collected in a peri-urban district, was found to be positive for Japanese encephalitis virus. Conclusions: The risk of flavivirus transmission in urban areas of Hanoi city due to the spread of *Culex* and *Aedes* mosquitoes could be facilitated by livestock keeping.

## 1. Introduction

Mosquitoes are considered the deadliest animal in the world, since they are vectors for many important human pathogens, causing a large number of human deaths per year. One group of such pathogens are viruses within the genus flavivirus (Flaviviridae family), and include some significant human pathogens such as dengue virus (DENV) and Japanese encephalitis virus (JEV), spread by the bites of mosquitoes in the genera *Aedes* and *Culex* [1,2]. 

JEV is transmitted by *Culex* species—primarily *Cx. tritaeniorhynchus*—and its transmission cycle involves water birds and pigs as maintenance and amplification hosts, respectively, while humans are dead-end hosts [3]. This virus is the leading cause of viral encephalitis in Asia, estimated to cause nearly 68,000 cases and more than 13,000 deaths annually [4]. The disease mostly affects children in rural rice-growing areas, which provide ideal mosquito breeding sites. There are no effective antiviral treatments [5]; immunization through vaccination is the only method for sustainable long-term prevention. In complement to vector control, vaccination has contributed to decreasing the risk of infection in endemic areas and is now part of routine immunization programs in several Asian countries [1], including Vietnam. 

DENV is transmitted primarily by *Ae. aegypti*, and secondarily by *Ae. albopictus* [6]. These mosquito vectors are adapted to peri-domestic urban habitats, where they breed in water storage containers. They bite during daytime; therefore, traditional vector control methods such as bed nets have limited effectiveness. Weather components such as rainfall and temperature have been shown to strongly impact the distribution, development, and survival of *Aedes* species [7]. As a result of climatic, demographic, and socioeconomic changes, their abundance and geographical range are expected to increase in the near future—especially *Ae. albopictus*—leading to more frequent and severe epidemics [8,9].

Urban livestock keeping is an important and integral part of cities in many tropical and low- and middle-income countries, where it not only ensures highly nutritious food in urban markets, but also provides urban inhabitants with livelihood options [10,11]. However, there are also risks associated with urban livestock keeping that should not be neglected, including zoonotic diseases that can be transferred from the animals to humans [12]. Animals such as pigs and cattle may increase the presence of mosquitoes, since they provide blood sources for them [13]. With a fast-growing human population and urbanization worldwide, humans, animals, and mosquitoes are in closer contact. This can increase the risk for humans to become infected with mosquito-borne diseases [14]. In Ethiopia and Pakistan, study findings indicate that the presence of cattle or goats near to homes tends to increase the human-biting rate of *Anopheles* mosquitoes [15,16]. Other studies also highlight that keeping goats and/or other medium-sized livestock—such as sheep and pigs—may contribute to increased risk of non-zoonotic mosquito-borne diseases, such as malaria, by increasing the number of vectors [17,18]. One study in Vietnam found that urban pig keeping was associated with an increase in the number of JEV vectors; in this study, Lindahl et al. indicated that the number of pigs in households was correlated with the increase in the numbers of *Cx. tritaeniorhynchus* and all of the mosquitoes in the *Cx. vishnui* subgroup, while the density of people in the households increased the number of *Cx. quinquefasciatus* [19]. In another study, collected *Culex* mosquitoes were found to be positive for JEV, belonging to genotypes I and III, while 100% of pigs in the study were found to have JEV antibodies [20]. JEV has been confirmed to circulate in pigs, mosquitoes, and humans in many parts of Vietnam, in spite of the high vaccination coverage [21].

Since 2008, after merging areas of surrounding provinces, Hanoi city—now called “Great Hanoi”—has been undergoing rapid urbanization. Great Hanoi is divided into 12 central urban and peripheral districts, 17 peri-urban districts, and 1 town; those districts have different livestock densities [22]. With those characteristics, along with a warm, humid, subtropical climate, Hanoi city is an endemic area of mosquito-borne diseases such as dengue fever and Japanese encephalitis [21]. 

However, there is limited research on the association between urban livestock keeping and mosquito population, both globally and in Vietnam. This paper explores how livestock keeping in and around a large city is associated with the presence of mosquitoes and the risk of them spreading flaviviruses.

## 2. Materials and Methods

### 2.1. Design

The entomological study was performed in parallel with a questionnaire survey assessing the knowledge and practices of households with and without livestock [23], conducted in Hanoi city from September to October 2018. 

### 2.2. Study Sites 

Six districts were selected based on livestock population: two central urban districts, which comprise the inner districts of old Hanoi, with no livestock—namely, **Ba Dinh and Cau Giay**—two peripheral districts that are expanded city districts of new Hanoi, with some livestock—namely, **Ha Dong and Bac Tu Liem**—and two peri-urban districts that comprise suburban districts with higher livestock populations, namely, **Chuong My and Dan Phuong** (Table 1). 

A household with livestock was defined as a household having at least one larger livestock species (pig, cattle, goat, or larger), or at least five smaller food-producing animals (chickens, rabbits, ducks, geese, etc.). The sample size and sampling were described clearly in our previous study [23]. In our study, 513 households were recruited to collect the livestock information and vector samples, as shown below.

### 2.3. Mosquito and Larva Collection and Identification 

Adult mosquitoes were collected close to animal farms (outdoors) as well as close to human habitats (indoors). Due to limited resources and the practical characteristics of each mosquito trap, only non-baited CDC light traps [24] were used outdoors, while both BG-Sentinel traps with lures [25] and non-baited CDC light traps were used indoors. In households without livestock, the outdoor collection was performed in the garden or yard. All containers in the households (pots, stagnant water in trash, ponds, etc.) were investigated with sweep nets to catch mosquito larvae. Before leaving the location, the traps were set and left overnight. Early the next morning, after 12–18 h, the traps were collected. Mosquito and larva samples were labelled by household ID and collection sites. The samples were put in a cool box and brought to the Vietnam National Institute of Hygiene and Epidemiology for storage at −80 °C to preserve virus integrity. The collected mosquitoes were counted and identified by species and gender, while larvae were also identified to genus level according to recommended keys [26]. 

### 2.4. Laboratory Techniques

The mosquito samples were grouped into 385 pools of a maximum of 48 individuals by species and collection site. Pools of mosquitoes were homogenized, and then total RNA was extracted using an RNeasy Mini Kit (Qiagen, Hilden, Germany), according to the manufacturer’s instructions, with the following modifications: The mosquitoes were homogenized in 2 mL tubes containing 500 µL of 1× Minimum Essential Medium (MEM) using a mini pestle. After homogenization in 1× MEM, 50 µL of the homogenate was added to 350 µL of RLT buffer (with added 2-mercaptoethanol). Subsequently, the mixture was centrifuged at 14,000 rpm for 2 min. The supernatant was carefully removed and added to equal amounts of 70% ethanol. The MEM homogenization of each pool was stored at −80 °C for further analyses. 

In order to detect flaviviruses in the collected mosquitoes, a pan-flavivirus qPCR was used following the protocol of Patel et al. [27], with two different forward primers and one reverse primer (Table 2), generating products of 266 base pairs (bp). As positive controls, RNAs extracted from a local DENV-1 strain (D7709, Vietnam) and the JEV Nakayama strain (genotype III) in dilutions of 10^−1^–10^−3^ were used. 

The QuantiTect SYBR^®^ Green RT-PCR kit was used according to the manufacturer´s instructions (Qiagen-204245, Germany). In each reaction, 2× QuantiTect master mix, 0.4 µM of each primer (see Table 2), 1× QuantiTect RT mix, RNase-free H_2_O, and 5 µL of RNA—creating a total volume of 25 µL—were included. The PCR amplification conditions were as follows: reverse transcription at 50 °C for 30 min, initial activation at 95 °C for 15 min, followed by 45 cycles of denaturation at 94 °C for 15 s, annealing at 60 °C for 30 s, and extension at 72 °C for 30 s. 

A sample was considered positive if both duplicated samples had a Ct value below 40 and had correct melting temperatures (see Table 3). The cutoff points was validated based on the protocol of Patel et al. [27] in our lab in Uppsala. Gel electrophoresis was carried out with these positive samples to confirm whether the correct product had been amplified in the pan-flavivirus qPCR. A 2% agarose gel was prepared with 1× TAE, agarose (Sigma-Aldrich Solutions, Darmstadt, Germany), and 1 × RedSafe. A 100 bp DNA ladder (Invitrogen, Karlsruhe, Germany) was used. The PCR product was mixed with 1 µL of dye, and 5 µL was loaded onto the gel. The samples were run at 100 V for approximately 30 min. 

### 2.5. Long Non-Coding RNA Sequencing

The total RNA from the mosquito pool positive for the pan-flavivirus real-time PCR was sent for long non-coding RNA sequencing (LncRNA-seq) at Novogene Hong Kong (https://en.novogene.com/ (accessed on 26 September 2021)). The RNA library preparation included ribosomal RNA depletion as RNA enrichment, followed by paired-end sequencing with 150 bp per read using the Illumina NovaSeq 6000 sequencing platform, performed at Novogene. The number of raw reads reached around 50 million reads. The data analysis pipeline was used as described in a previous study [28]. Briefly, the raw clean reads were quality-trimmed using Trimmomatic v.0.36 (https://github.com/timflutre/trimmomatic (accessed on 26 September 2021)), followed by screening using Diamond v.0.9.15.116 (https://github.com/bbuchfink/diamond (accessed on 26 September 2021)) and BLASTn v.2.6.0+ against the complete NCBI non-redundant nucleotide and protein databases, with 1 × 10^−5^ as a cutoff e-value. Sequence reads that were indicated to match virus genomes were pulled out from the library by using seqtk/1.2 (https://github.com/lh3/seqtk (accessed on 26 September 2021)). These virus reads were then screened to extract the flavivirus sequences for further analyses.

### 2.6. Phylogenetic Analysis 

The flaviviruses sequences were extracted for phylogenetic inference. The reference genome of JEV was downloaded from GenBank (https://www.ncbi.nlm.nih.gov/genbank/ (accessed on 26 September 2021)) using “Japanese encephalitis virus” and “Vietnam” as search criteria. The dataset included 39 JEV sequences from Vietnam. All of the sequences were aligned using MAFTT [29]. The phylogenetic analysis was performed by using the IQ Tree web server, where a substitution model was selected first, followed by a pairwise maximum likelihood method for the tree inference [30]. We also included Usutu virus (NC_006551) as an outgroup for the analysis.

### 2.7. Data Analysis

Excel was used for data entry, and data were transferred to STATA 15.0 for management and analysis. The discrete variables related to the quantity of mosquitoes and larvae were described by the median interquartile range (IQR), as they did not follow a normal distribution. The Kolmogorov–Smirnov test was used to assess the normal distribution of variables. Spearman’s rho was used to describe the correlation between the quantity of each of the types of mosquitoes and larvae with the numbers of pigs, cattle, and poultry kept. The number of mosquitoes collected was over-dispersed (variance was much higher than the mean), with excessive zero value; therefore, a zero-inflated negative binomial regression model was built to identify the factors associated with the size of the mosquito population. In the models, the total number of mosquitoes and the number of *Culex* mosquitoes were set as the outcome variables, since the other collected mosquito species were too few in number. The possible risk factors included the district lived in, the number of people living in the households, the practice score of household respondents, the numbers of pigs, cattle, and poultry kept, and livestock presence. In this study, practice score was derived from the dataset of our previous survey [23]. The practice score ranged from 0 to 11 points and was based on the sum of all preventive methods that the household respondents had used to prevent and control mosquito-borne diseases (MBDs). Household respondents who acquired higher scores were presumed to have better levels of practice. Manual stepwise backward deletion of non-significant variables was applied to build the final models. A *p*-value < 0.05 was regarded as significant.

## 3. Results

In total, 12,861 mosquitoes and 2427 larva samples were collected from 513 households. Notably, in 103 households, no mosquitoes at all were collected. 

The distribution of mosquitoes and larvae in the six districts is described in Table 4 and Table 5 below. *Culex* was the most dominant genus, comprising more than 93% of the total collected adult mosquitoes. Among these, *Cx. tritaeniorhynchus* made up the majority, with 67.18%, followed by *Cx. gelidus, Cx. quinquefasciatus*, and *Cx vishnui* with approximately equal proportions of 8%. *Aedes* constituted the least abundant mosquitoes, with less than 1% amongst five genera. More *Ae. albopictus* were collected than *Ae. aegypti*. Other genera—including *Mansonia, Armigeres* and *Anopheles*—ranged from 1.21% to 3.82%.

Most mosquitoes were collected in peripheral districts, while fewer mosquitoes were found in the central urban districts—especially in Ba Dinh. The peripheral district of Ha Dong had the highest number of mosquitoes collected, followed by the peri-urban district of Dan Phuong. *Cx. tritaeniorhynchus* accounted for the highest proportion in peri-urban and peripheral districts, while *Cx. quinquefasciatus* was the dominant species in the central districts. *Ae. aegypti* were found mainly in central districts, whereas they could not be found in peripheral districts. *Ae. albopictus* appeared in peri-urban and peripheral districts much more often than in central districts. In central districts, *Armigeres* and *Anopheles* were not found, whereas only one *Mansonia* mosquito was found there. 

*Ae. albopictus* were the most common larvae collected, with more than 80% of the total number of larvae collected, while the percentage of *Ae. aegypti* larvae was 2.84%. Most of the *Ae. albopictus* larvae were collected in the peri-urban districts, followed by peripheral districts. *Ae. aegypti* larvae were mainly collected in central districts; however, their quantity was still lower than that of *Ae. albopictus* larvae. A total of 12.2% of the larvae collected belonged to the *Culex* species and were mostly found in the peripheral district of Ha Dong and in the peri-urban district of Chuong My. Notably, only one *Culex* larva was collected in Dan Phuong district, and no *Culex* larvae were detected in Bac Tu Liem district. There were 18 *Armigeres* larvae found in total—all from the peripheral district of Bac Tu Liem. Only one *Anopheles* larva was collected in Chuong My district. 

In general, more mosquitoes (both indoor and outdoor) were trapped in households with livestock than in households without livestock (Table 6). The medians of total mosquitoes, mosquitoes collected indoors, and mosquitoes collected outdoors in households with livestock were 9 ± 30 (median ± IQR), 3 ± 11, and 3 ± 16, respectively. Meanwhile, in households without livestock, the medians of total mosquitoes, mosquitoes collected indoors, and mosquitoes collected outdoors were 2 ± 5, 1 ± 3, and 0 ± 2, respectively. Similarly, households with pigs had much more total mosquitoes, collected indoors and outdoors, than households without pigs, while there was only a minor difference in the median numbers of total collected mosquitoes, mosquitoes collected indoors, and mosquitoes collected outdoors between the households with and without cattle. The total numbers of mosquitoes caught in households with poultry were higher than in households without poultry; however, the median numbers of mosquitoes collected indoors, and outdoors were similar in households with and without poultry. 

Regarding mosquito species, as described above, *Culex* was the most commonly collected species, of which the number of total *Culex* mosquitoes was higher in households with livestock, pig keeping, cattle keeping, and poultry keeping, as compared to households without livestock, pig keeping, cattle keeping, and poultry keeping. Nevertheless, in terms of the numbers of mosquitos collected indoors and outdoors, the median numbers of *Culex* mosquitoes collected were only slightly higher or equal between the households with and without livestock, pig keeping, cattle keeping, and poultry keeping. There were no significant differences in the median numbers of total mosquitoes, mosquitoes collected indoors, and mosquitoes collected outdoors of other species and total larvae species collected in households with and without livestock, pig keeping, cattle keeping, and poultry keeping.

The numbers above represent the median ± IQR values, while the numbers below represent the min–max values.

Table 7 presents the correlation of the numbers of mosquitoes and larvae collected with the numbers of livestock kept at the households. Spearman’s rho test showed that the number of pigs kept was positively correlated with the numbers of total mosquitoes, total mosquitoes indoors, total mosquitoes outdoors, total *Culex*, *Culex* indoors, *Culex* outdoors, total *Anopheles*, *Anopheles* indoors, *Anopheles* outdoors, total *Armigeres*, *Armigeres* outdoors, total *Mansonia*, *Mansonia* indoors, and *Mansonia* outdoors.

No positive correlation was found between the numbers of mosquitoes and larvae and the numbers of cattle kept. While the test indicated a weak positive correlation between total *Anopheles*, *Anopheles* indoors, and *Anopheles* outdoors, all larval species were positively correlated with the quantity of poultry. 

Table 8 shows the final models for factors associated with the collected numbers of mosquitoes in general and *Culex* mosquitoes in particular. In both models, the factor of livestock keeping in the part of the logit model predicting excessive zeros was not statistically significant (*p* > 0.05). The expected log-change in the number of mosquitoes was 0.01 for a one-unit increase in pig keeping (*p* < 0.001) and −0.09 for one-unit increase in the practice scores of household respondents (*p* < 0.05). In the first model, the peri-urban districts had an expected log number of mosquitoes of 1.1 lower than peripheral districts and 1.5 higher than central urban districts (*p* < 0.001). In the second model, the peri-urban districts had an expected log number of mosquitoes of 1.2 lower than peripheral districts and 1.4 higher than central urban districts (*p* < 0.001).

### 3.1. Pan-Flavivirus q-PCR and Gel Electrophoresis 

In total of 385 pools, there were 248 pools of *Cx. tritaeniorhynchus*, 34 pools of *Cx. vishnui* and *pseudovishnui*, 34 pools of *Cx. quinquefasciatus*, 32 pools of *Cx. gelidus*, 1 pool of *Cx. fuscocephala*, 17 pools of *Anopheles*, 6 pools of *Armigeres*, 7 pools of *Mansonia*, and 6 pools of *Aedes*. Only one pool from Hong Phong commune, Chuong My district—consisting of *Cx. tritaeniorhynchus*—showed a suspected positive result, with a Ct value of 38.27, and was therefore subjected to gel electrophoresis. This pooled sample showed a band in the gel electrophoresis along with the positive controls, as shown in Figure 1.

### 3.2. Sequencing Results

We obtained 116,961,210 clean reads, with an efficiency of 97.82%, after removing the low-quality reads and adaptors. The paired-end sequencing read 1 (R1) and read 2 (R2) were combined, and the reads were mapped to the database (NCBI non-redundant nucleotide and protein database). Two reads matched to the poly protein of JEV strain SXYC 1523 (GenBank accession No. ARX98191 for the amino acid and KY078829 for the nucleotide sequences, respectively). The 216 bp JEV sequence was extracted and included in the JEV dataset. The phylogenetic tree showed that the JEV sequence recovered from the mosquito pool belonged to genotype I (GT-I) (Figure 2); however, the sequence was too short for further analysis.

## 4. Discussion

### 4.1. Mosquito Collection Method

In our study, BG-Sentinel traps (with lures) were used indoors only, because they need to connect with a power source. This kind of trap is attractive for *Ae. aegypti, Ae. albopictus, Cx. quinquefasciatus*, and other selected species [25]. Non-baited CDC light traps were more flexible to use both indoors and outdoors, and can catch a wide range of mosquito genera, including *Culex* and *Aedes* species [31]. Therefore, we used more CDC light traps to collect mosquitoes in this study. The other methods that could have been used to collect mosquitoes were backpack aspirators or gravid traps. We previously utilized backpack aspirators for collecting mosquitoes in another study regarding livestock keeping and dengue. However, not many *Aedes* mosquitoes were collected, perhaps because this method was not optimal due to the time difference in collection between households over the course of a day [32]. Gravid traps were deemed to have too narrow a spectrum in catching mosquitoes, and could easliy be in the way in the household areas. Hence, along with the available logistic arrangement, we decided to use the combination of CDC light traps and BG-Sentinel traps.

### 4.2. Mosquito Distribution and Composition

There were five genera of mosquitoes collected at the study sites of Hanoi city, including *Aedes, Culex, Mansonia, Anopheles*, *and Armigeres*. These genera were also found in a previous study conducted in eight provinces in Vietnam [33], as well as in a city in India [34]. The genus *Culex* was most abundant in these studies, as well as in ours. This finding was also similar to those of other entomological studies in Vietnam [35,36]; Pakistan [37]; Guwahati city, India [38]; Melbourne, Australia [39]; Atlanta, GA, USA [40]; Mexico City, Mexico [41]; and Palangka Raya City, Indonesia [42]. These results indicate a wide dispersion of *Culex* mosquitoes in tropical and subtropical areas. 

*Aedes* mosquitoes were the least collected as compared to the other mosquito species in our entomological survey, consistent with the findings of an earlier study in Laos [43]; nevertheless, it is remarkable that their larvae—especially *Ae. albopictus*—were more commonly detected than those of the other species. An aggressive spread of *Ae. aegypti* and *Ae. albopictus* has been recorded across the globe [9,44]. These two *Aedes* species, which constitute the primary and secondary vectors of dengue fever, were found in all districts in our study. This finding implies that there is a risk for people living in both urban and suburban areas of Hanoi city to become infected with DENV. A previous study in Khanh Hoa province, Vietnam, also proved that people in rural areas had at least an equal risk of catching dengue fever compared to people in urban areas [45]. 

### 4.3. Risk factors of Vector Presence 

Our study indicated that the district lived in, pig keeping, and preventive practices of people were associated with the abundance of mosquito populations—especially *Culex* mosquitoes. Compared to peri-urban districts, central urban districts had fewer mosquitoes, whereas the peripheral districts had much more mosquitoes. A possible explanation for this finding might be related to the activity of livestock keeping in general, and pig rearing in particular. Rearing of pigs and other livestock were common agricultural activities of all areas in Hanoi city in the past; however, with the rapid urbanization and industrialization of the capital city, this practice has recently been restricted in the central urban areas [22]. This may explain why fewer mosquitoes were found in these areas as compared to peripheral and peri-urban districts where keeping livestock is still common. Environmental factors could also contribute to the differences in the presence of mosquitoes between the three categories of area. In our case, the domestic wastewater and sewer systems—which are important breeding sites of many mosquito species, such as *Aedes* [46,47] and *Culex* [40]—are more developed in the central urban districts as compared to the peri-urban districts. Meanwhile, peripheral districts are in a mixed situation, between central urban and peri-urban districts, with considerable speed of urbanization, while retaining the habit of keeping livestock. The development in these areas has created more breeding grounds for mosquitoes, such as stagnant water at construction sites, abandoned houses, or water feeding and cleaning for livestock-keeping activities amidst residential areas, leading to the increased possibility of mosquito population growth in these areas. In our study, a majority of the mosquitoes were collected in these peripheral districts, followed by the peri-urban districts. Our previous study showed that people living in central urban districts had better knowledge about MBDs than people living in peri-urban districts, and people with better knowledge about MBDs had better practices against MBDs [23]. This could also explain why we found fewer mosquitoes in the central urban districts than in the peri-urban districts. 

Rearing of pigs can potentially provide blood meals for *Culex* mosquitoes, and pigs act as amplifying hosts for JEV [1]. Our study concluded that pig keeping increased the number of mosquitoes—especially *Culex* mosquitoes. This is further convincing evidence to support the previous study that was conducted in an urban area of southern Vietnam [19]; in that study, Lindahl et al. found that there was a strong association between rearing of pigs and the increased presence of the *Cx. tritaeniorhynchus* population—the main vector of JEV. Hence, communication programs from the health sector and the veterinary sector need to be implemented, in order to enhance the knowledge and practice of pig farmers with regard to JEV prevention and control. In our study, cattle keeping was not correlated with the number of mosquitoes, although it may contribute to higher numbers of *Anopheles* mosquitoes [17,42]. However, in some cases, the keeping of cattle, or ruminants such as goats and sheep, is used as a means of zooprophylaxis in some malaria-endemic countries [16]. Poultry keeping was correlated with the number of Anopheles mosquitoes and various kinds of larvae. To our knowledge, no evidence has been found to demonstrate the relationship between poultry keeping and the growth of any species of mosquito. On the other hand, a smaller study by Jakobsen et al. [32] also could not find any association between livestock keeping—including pig-keeping, poultry-keeping, and the keeping of ruminants—and the presence of *Aedes* mosquitoes. Therefore, further research should be conducted in order to explore these potential associations.

Our study found that a better practice score reduced the number of mosquitoes. This finding was expected, as the abundance of mosquitoes will be decreased when people more often use preventive practices against mosquito growth. For instance, in a study conducted in Ho Chi Minh City, Vietnam, the results indicated that appropriate cover of containers diminished the risk of the presence of *Ae. aegypti* larvae as compared with those with inappropriate cover [48]. Similarly, people in suburban Washington who practiced source-reduction had lower numbers of *Ae. albopictus* and *Cx. pipiens* pupae in the containers [49]. Moreover, in Thailand, Rahman et al. [50] revealed that lack of dengue-preventive practices was significantly associated with higher abundance of adult female and immature *Ae. aegypti*. As shown in our previous studies, the preventive practices of households with and without livestock in the study sites was not good [23], and people often felt powerless with regards to mosquito prevention or perceived it as somebody else’s responsibility [51]. Therefore, it is important to improve the practices related to larval and mosquito control of the people in order to minimize the transmission of MDBs in Hanoi city.

### 4.4. Laboratory Results

The laboratory analyses detected flavivirus in one *Cx. tritaeniorhynchus* pool. Thereafter, we attempted to recover the sequences from the initial qPCR screening, but without success. By using RNA-Seq, we finally found two reads mapped to JEV genotype I, which was also consistent with our qPCR results. However, we only obtained two reads from over 100 million, suggesting that the virus load in the *Cx. tritaeniorhynchus* mosquito is very low. Combined with all of our results, this mosquito pool most likely carried JEV genotype I, which is one of the dominant genotypes in Asia [52,53,54], although our positive control of JEV belonged to genotype III. Moreover, this result was expected, since this mosquito pool was from the Hong Phong commune, Chuong My district, where one JEV case was recorded in 2019 (unpublished data). In one study conducted earlier in the Ha Tay province (now belonging to Hanoi city), JEV was isolated from two pools of *Cx. tritaeniorhynchus* [33]. Hence, this study shows an existing risk of JEV transmission in Hanoi city—especially in the peri-urban and peripheral districts, with the high abundance of the primary vector of JEV. It is recommended to focus on effective vector control measures, as well as an increase in the immunization rate via JEV vaccination (both primary and booster doses)—at least for the most vulnerable group of children under 15 years old in the endemic area.

### 4.5. Strengths and Weaknesses of the Study 

The strengths of this study were the combination of BG-Sentinel traps and CDC light traps, which can catch a variety of different mosquito species, and the application of a zero-inflated statistical model to determine the associated factors that were appropriate for data analysis, since a number of households did not render any mosquitoes, leading to many zero values in the variable of mosquito quantity. In addition, the combination with a survey on knowledge and practices allowed us to use a practice score to see the influence on mosquito numbers.

Nevertheless, our study still has several limitations regarding the collection of vectors and generalizability. Firstly, the time of data collection was at the end of the vector season. Secondly, the operation time of the traps varied slightly, in spite of attempts to standardize. These elements affected the quantity of collected mosquitoes. In addition, this study’s findings may not be generalizable to other parts of Vietnam or other countries, as our study sites were categorized into three different settings, with the classified standard based on numbers of livestock kept.

## 5. Conclusions

The distribution of the vector population in Hanoi city was heterogeneous, and livestock play a role in influencing this. Adult mosquitoes were trapped mostly in peripheral districts, followed by peri-urban districts. However, the number of larvae was much higher in peri-urban districts as compared to peripheral ones. The lowest numbers of mosquitoes and larvae were found in central urban districts. The *Culex* genus made up the most abundant adult mosquitoes amongst the five different genera collected. The highest percentage of mosquito species was *Cx. tritaeniorhynchus*, followed by other *Culex* species. Adult *Aedes* mosquitoes were less commonly collected, but their larvae accounted for the majority. Our results indicate that there was a positive association between pig rearing and the size of the mosquito population—mainly *Culex* mosquito species. Households with better preventive practices reduced the possibility of mosquito presence. Our laboratory analysis revealed only one JEV-positive pool of *Cx. tritaeniorhynchus*, indicating a low infection rate. Our findings, however, suggest that vector control measures should be applied in all areas of Hanoi city, taking livestock keeping into account, in order to decrease the vector population and, thus, prevent the risk of transmission of flaviviruses such as DENV and JEV. The need for increasing JEV vaccine coverage for the vulnerable groups in the endemic areas should also be emphasized, even in urban provinces. 

## Figures and Tables

**Figure 1 viruses-13-02291-f001:**
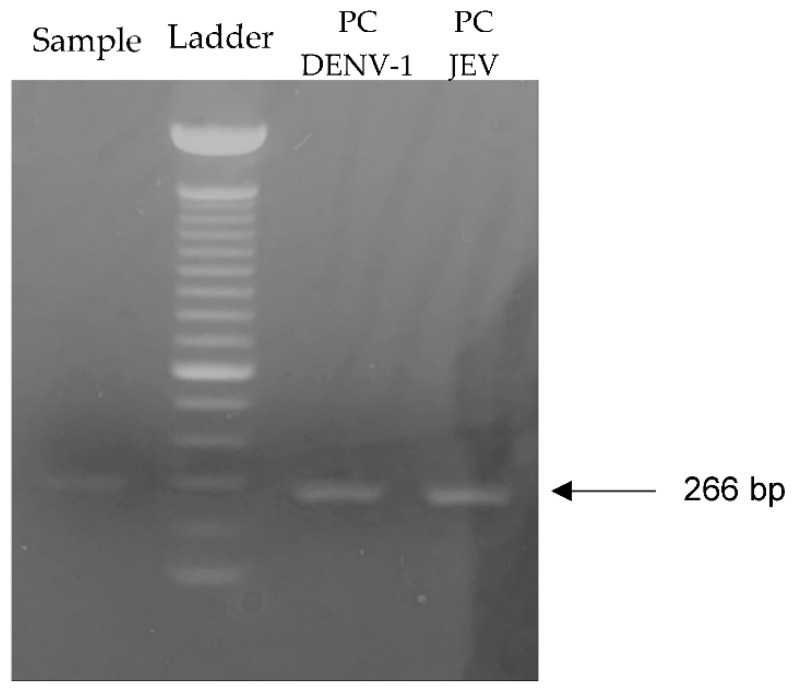
PCR result of the positive pool of *Culex tritaeniorhynchus.* The positive *Culex tritaeniorhynchus* sample was obtained from pool 6. PC DENV-1 and JEV represented the dengue virus serotype 1 and Japanese encephalitis virus that were used as positive controls. Both positive controls have the same band length of 266 bp.

**Figure 2 viruses-13-02291-f002:**
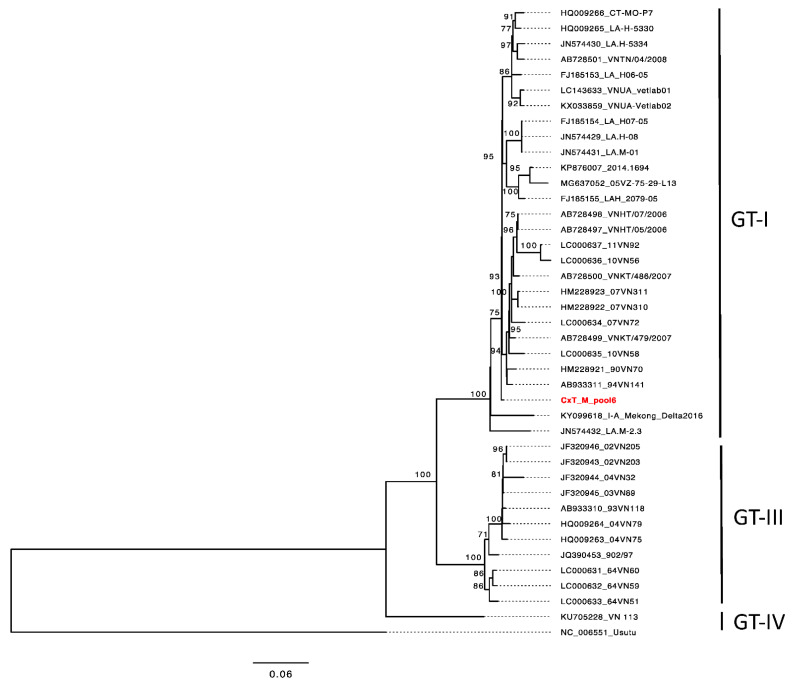
Phylogenetic tree of JEV. The tree was constructed via the maximum-likelihood method implemented in the IQ tree server, using the best fit GTR + G + r substitution model. The ultrafast bootstrap values > 75 are shown on the branches. The tree was rooted on Usutu virus Vienna 2001 strain (GenBank accession no. NC_006551.1). The scale bar shows the nucleotide substitutions per site. The highlight taxa in the tree show the JEV sequences recovered from 385 pools in this study. Genotypes (GTs) I–IV are shown to the right.

**Table 1 viruses-13-02291-t001:** The number of livestock-keeping households and non-livestock-keeping households included in the survey in Hanoi city, Vietnam.

	Households with Livestock	Households without Livestock
Central districts	0	102
Peripheral districts	104	98
Peri-urban districts	129	80
**Total**	**233**	**280**

**Table 2 viruses-13-02291-t002:** Names and sequences of the primers used in the pan-flavivirus qPCR.

*Primer*	*Sequence*
*Flavi all S*	TACAACATgATggggAARAgAgARAA
*Flavi all S2*	TACAACATgATgggMAAACgYgARAA
*Flavi all AS4*	gTgTCCCAGCCNgCKgTRTCRTC

**Table 3 viruses-13-02291-t003:** Melting temperature of different flaviviruses generated during the development of the pan-flavivirus qPCR protocol.

*Virus Type*	*Melting Temperature (°C)*
West Nile virus	79.0
Zika virus	81.0
DENV1	79.0
DENV2	81.0
DENV3	80.5
DENV4	80.5
JEV	81.5
Yellow fever virus	81.5
Negative control (primer)	74.5

**Table 4 viruses-13-02291-t004:** Distribution of adult mosquitoes by collection site.

Mosquito Species	Periurban Districts				Peripheral Districts				Central Districts				All	
	*Chuong My*		*Dan Phuong*		*Bac Tu Liem*		*Ha Dong*		*Ba Dinh*		*Cau Giay*			
	*n*	*%*	*n*	*%*	*n*	*%*	*n*	*%*	*n*	*%*	*n*	*%*	*n*	*%*
*Aedes aegypti*	1	0.08	2	0.09	0	0	0	0	3	5.9	5	2.7	**11**	**0.085**
*Aedes albopictus*	15	1.15	30	1.38	25	1.58	12	0.16	3	5.9	1	0.5	**86**	**0.67**
Other *Aedes* sp.	1	0.08	0	0	0	0	0	0	1	1.9	1	0.5	**3**	**0.023**
*Culex tritaeniorhynchus*	798	61.05	1578	72.2	1020	64.52	5237	69.37	3	5.9	5	2.7	**8641**	**67.18**
*Culex vishnui*	31	2.37	231	10.57	213	13.47	535	7.09	0	0	4	2.2	**1014**	**7.89**
*Culex pseudovishnui*	0	0	4	0.18	0	0	4	0.05	0	0	0	0	**8**	**0.062**
*Culex quinquefasciatus*	106	8.11	88	4.03	127	8.03	589	7.8	41	80.4	170	90.4	**1121**	**8.72**
*Culex gelidus*	91	6.96	71	3.25	116	7.34	878	11.63	0	0	1	0.5	**1157**	**8.99**
*Culex fuscocephala*	22	1.68	0	0	0	0	0	0	0	0	0	0	**22**	**0.17**
*Mansonia* sp.	31	2.37	41	1.88	20	1.27	63	0.84	0	0	1	0.5	**156**	**1.21**
*Armigeres* sp.	9	0.68	21	0.96	7	0.44	114	1.51	0	0	0	0	**151**	**1.18**
*Anopheles* sp.	202	15.45	119	5.46	53	3.35	117	1.55	0	0	0	0	**491**	**3.82**
** *Total* **	**1307**	**100**	**2185**	**100**	**1581**	**100**	**7549**	**100**	**51**	**100**	**188**	**100**	**12,861**	**100**

**Table 5 viruses-13-02291-t005:** Distribution of larvae species by collection site.

Larval Species	*Peri-Urban Districts*				*Peripheral Districts*				*Central Districts*				*All*	
	*Chuong My*		*Dan Phuong*		*Bac Tu Liem*		*Ha Dong*		*Ba Dinh*		*Cau Giay*			
	*n*	%	*n*	%	*n*	%	*n*	%	*n*	%	*n*	%	*n*	%
*Aedes aegypti*	0	0	1	0.4	0	0	0	0	45	23.2	23	57.5	**69**	**2.84**
*Aedes albopictus*	1100	88.36	233	99.2	156	89.7	388	72	149	76.8	17	42.5	**2043**	**84.18**
*Culex* sp.	144	11.56	1	0.4	0	0	151	28	0	0	0	0	**296**	**12.2**
*Armigeres* sp.	0	0	0	0	18	10.3	0	0	0	0	0	0	**18**	**0.74**
*Anopheles* sp.	1	0.08	0	0	0	0	0	0	0	0	0	0	**1**	**0.04**
** *Total* **	**1245**	**100**	**235**	**100**	**174**	**100**	**539**	**100**	**194**	**100**	**40**	**100**	**2427**	**100**

**Table 6 viruses-13-02291-t006:** Numbers of mosquitoes/larvae in the households in Hanoi city.

	Households with Livestock *n* = 233	Households without Livestock n = 280	Households with Pigs *n* = 211	Households without Pigs *n* = 302	Households with Cattle *n* = 13	Households without Cattle *n* = 500	Households with Poultry *n* = 69	Households without Poultry *n* = 444
Total mosquitoes ***Median* ± *IQR*** ***Min–max***	9 ± 30 0–1385	2 ± 5 0–342	9 ± 34 0–1385	2 ± 6 0–479	5 ± 46 0–934	4 ± 13 0–1385	7 ± 18 0–142	3 ± 13 0–1385
Total mosquitoes indoors	3 ± 11 0–860	1 ± 3 0–124	3 ± 12 0–860	1 ± 3 0–124	1 ± 3 0–860	1 ± 6 0–176	1 ± 5 0–860	1 ± 6 0–27
Total mosquitoes outdoors	3 ± 16 0–1370	0 ± 2 0–337	3 ± 18 0–1370	0 ± 2 0–478	2 ± 44 0–1370	1 ± 5 0–478	1 ± 11 0–141	1 ± 5 0–1370
Total *Aedes*	0 ± 0 0–4	0 ± 0 0–4	0 ± 0 0–4	0 ± 0 0–4	0 ± 0 0–0	0 ± 0 0–4	0 ± 0 0–1	0 ± 0 0–4
*Aedes* indoors	0 ± 0 0–4	0 ± 0 0–4	0 ± 0 0–3	0 ± 0 0–4	0 ± 0 0–0	0 ± 0 0–4	0 ± 0 0–1	0 ± 0 0–4
*Aedes* outdoors	0 ± 0 0–2	0 ± 0 0–2	0 ± 0 0–2	0 ± 0 0–2	0 ± 0 0–0	0 ± 0 0–2	0 ± 0 0–1	0 ± 0 0–2
Total *Culex*	7 ± 25 0–1345	2 ± 5 0–341	8 ± 28 0–1345	2 ± 6 0–462	5 ± 43 0–1345	3 ± 12 0–932	6 ± 16 0–142	3 ± 11 0–1345
*Culex* indoors	2 ± 10 0–858	1 ± 2 0–124	2 ± 10 0–858	1 ± 3 0–124	1 ± 3 0–858	1 ± 5 0–171	1 ± 4 0–26	1 ± 5 0–858
*Culex* outdoors	2 ± 14 0–1341	0 ± 2 0–336	2 ± 17 0–1341	0 ± 2 0–462	2 ± 41 0–462	1 ± 5 0–1341	1 ± 10 0–141	0 ± 5 0 -1341
Total *Anopheles*	0 ± 1 0–48	0 ± 0 0–33	0 ± 1 0–48	0 ± 0 0–33	0 ± 2 0–17	0 ± 0 0–48	0 ± 2 0–12	0 ± 0 0–48
*Anopheles* indoors	0 ± 1 0–7	0 ± 0 0–3	0 ± 0 0–7	0 ± 0 0–3	0 ± 0 0–2	0 ± 0 0–7	0 ± 1 0–3	0 ± 0 0–7
*Anopheles* outdoors	0 ± 0 0–41	0 ± 0 0–32	0 ± 0 0–41	0 ± 0 0–32	0 ± 1 0–16	0 ± 0 0–41	0 ± 1 0–12	0 ± 0 0–41
Total *Armigeres*	0 ± 0 0–28	0 ± 0 0–11	0 ± 0 0–28	0 ± 0 0–12	0 ± 0 0–12	0 ± 0 0–28	0 ± 0 0–2	0 ± 0 0–28
*Armigeres* indoors	0 ± 0 0–25	0 ± 0 0–11	0 ± 0 0–25	0 ± 0 0–11	0 ± 0 0–0	0 ± 0 0–25	0 ± 0 0–2	0 ± 0 0–25
*Armigeres* outdoors	0 ± 0 0–12	0 ± 0 0–2	0 ± 0 0–11	0 ± 0 0–12	0 ± 0 0–12	0 ± 0 0–11	0 ± 0 0–1	0 ± 0 0–12
Total *Mansonia*	0 ± 0 0–23	0 ± 0 0–7	0 ± 0 0–23	0 ± 0 0–7	0 ± 0 0–0	0 ± 0 0–23	0 ± 0 0–2	0 ± 0 0–23
*Mansonia* indoors	0 ± 0 0–10	0 ± 0 0–4	0 ± 0 0–10	0 ± 0 0–4	0 ± 0 0–0	0 ± 0 0–10	0 ± 0 0–1	0 ± 0 0–10
*Mansonia* outdoors	0 ± 0 0–19	0 ± 0 0–3	0 ± 0 0–19	0 ± 0 0–3	0 ± 0 0–0	0 ± 0 0–19	0 ± 0 0–2	0 ± 0 0–19
*Aedes* larvae	0 ± 0 0–261	0 ± 0 0–100	0 ± 0 0–261	0 ± 0 0–100	0 ± 0 0–52	0 ± 0 0–261	0 ± 0 0–185	0 ± 0 0–261
*Culex* larvae	0 ± 0 0–140	0 ± 0 0–7	0 ± 0 0–140	0 ± 0 0–14	0 ± 0 0–0	0 ± 0 0–140	0 ± 0 0–140	0 ± 0 0–14
*Armigeres* larvae	0 ± 0 0–12	0 ± 0 0–0	0 ± 0 0–12	0 ± 0 0–0	0 ± 0 0–0	0 ± 0 0–12	0 ± 0 0–12	0 ± 0 0–0
*Anopheles* larvae	0 ± 0 0–1	0 ± 0 0–0	0 ± 0 0–1	0 ± 0 0–0	0 ± 0 0–0	0 ± 0 0–1	0 ± 0 0–1	0 ± 0 0–0

**Table 7 viruses-13-02291-t007:** Correlation between the quantity of mosquitoes/larvae and the numbers of livestock kept.

	*Numbers of Pigs*		*Numbers of Cattle*		*Numbers of Poultry*	
	Spearman’rho	*p*-Value	Spearman’rho	*p*-Value	Spearman’rho	*p*-Value
**Total mosquitoes**	**0.34**	**<0.001**	0.04	0.35	0.08	0.06
Total mosquitoes indoors	**0.23**	**<0.001**	0.008	0.84	−0.03	0.47
Total mosquitoes outdoors	**0.29**	**<0.001**	0.07	0.07	0.07	0.07
**Total *Aedes***	0.02	0.59	−0.07	0.09	−0.04	0.27
*Aedes* indoors	0.01	0.76	−0.06	0.17	−0.06	0.14
*Aedes* outdoors	0.02	0.52	−0.04	0.35	0.005	0.9
**Total *Culex***	**0.34**	**<0.001**	0.05	0.26	0.07	0.09
*Culex* indoors	**0.22**	**<0.001**	0.01	0.79	−0.02	0.56
*Culex* outdoors	**0.28**	**<0.001**	0.08	0.054	0.07	0.1
**Total *Anopheles***	**0.33**	**<0.001**	0.07	0.1	**0.2**	**<0.001**
*Anopheles* indoors	**0.26**	**<0.001**	0.06	0.15	**0.15**	**<0.001**
*Anopheles* outdoors	**0.24**	**<0.001**	0.06	0.12	**0.14**	**0.001**
**Total *Armigeres***	**0.09**	**0.036**	0.03	0.48	−0.02	0.64
*Armigeres* indoors	0.07	0.11	−0.03	0.37	−0.007	0.86
*Armigeres* outdoors	**0.1**	**0.016**	0.06	0.14	−0.04	0.3
**Total *Mansonia***	**0.2**	**<0.001**	−0.06	0.13	0.03	0.46
*Mansonia* indoors	**0.12**	**0.006**	−0.04	0.27	−0.04	0.3
*Mansonia* outdoors	**0.19**	**<0.001**	−0.05	0.24	0.04	0.28
***Aedes* larvae**	0.04	0.37	0.005	0.89	**0.14**	**0.001**
***Culex* larvae**	0.06	0.16	−0.02	0.58	**0.15**	**<0.001**
***Armigeres* larvae**	0.07	0.09	−0.01	0.8	**0.15**	**<0.001**
***Anopheles* larvae**	0.06	0.16	−0.007	0.86	**0.12**	**<0.01**

**Table 8 viruses-13-02291-t008:** Factors associated with the quantity of mosquitoes using a zero-inflated negative binomial regression model with the Vuong test.

	Coefficient	CI 95%	*p*-Value
**Model 1: Total number of all mosquito species as the dependent variable**			
** *District (ref—Peri urban)* **			
Peripheral	1.1	0.76–1.42	**<0.001**
Central urban	−1.5	−1.9–(−1.1)	**<0.001**
** *Number of pigs kept* **	0.01	0.007–0.02	**<0.001**
** *Practice score ** **	−0.09	−0.17–(−0.02)	**0.013**
**Zero-inflated variable** ** *Livestock keeping* **	−7	−390,260.8–390,246.7	>0.05
**Model 2: Total number of Culex mosquitoes as the dependent variable**			
** *District (ref—Peri urban)* **			
Peripheral	1.2	0.85–1.54	**<0.001**
Central urban	−1.4	−1.9–(−0.98)	**<0.001**
** *Number of pigs kept* **	0.01	0.007–0.02	**<0.001**
** *Practice score ** **	−0.09	−0.17–(−0.01)	**0.019**
**Zero-inflated variable** ** *Livestock keeping* **	13.2	−286,038.3–286,064.7	>0.05

* Data from a previous study [23]; CI: confidence interval; Ref: reference.

## Data Availability

The data presented in this study are available on request from the corresponding author.

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
