# Peer review of "The Distribution and Composition of Vector Abundance in Hanoi City, Vietnam: Association with Livestock Keeping and Flavivirus Detection"

_viruses, 2021, doi:10.3390/v13112291_

Round 1

Reviewer 1 Report

The distribution and composition of vector abundance in Hanoi city, Vietnam: its association with livestock keeping and flavivirus detection.

Please check all the virus and mosquito’s genus - Italic letter

Page 1 line 30: Tritaeniorhynchus - tritaeniorhynchus

Table: Please check all the tables. They are difficult to read and understand. The abbreviation used in the table needs to be explained.

Introduction

More information about JEV infection in humans and the pig population in Vietnam is needed.

Page 2 line 75-76: One study in Vietnam found that urban pig keeping was associated with the increasing number of JEV vector [19].

Please add more detail about this finding.

Discussion

Page 4 line 381-382: Aedes mosquitoes were the least collected as compared to the other mosquito species in our entomological survey, which was similar to an earlier study in Lao PDR [39]. Nevertheless, it is remarkable that its larvae, especially Ae. albopictus, were more commonly detected than the other species.

            Please discuss more about the adult mosquito collection method used in this study. Are there other methods that are more suitable for collecting Aedes mosquitoes? Vacuum?

Round 2

Reviewer 2 Report

Please, see the attachment for comments and suggestions.

Thanks.

Author Response

  • Line 160-161:

Please, (Invitrogen, Karlsruhe, Germany) should come immediately after DNA ladder. So, the statement should read ‘A 100 bp DNA ladder (Invitrogen, Karlsruhe, Germany) was used.

Response: Thank you. We already changed the order of those words.

  • Table 4 (Line 242-243):

The punctuation after Culex (Culex. pseudovishnui) in the seventh row should be deleted.

Response: Thank you. We deleted it.

  • Table 6 (Line 278):

For conformity, I will not put asterisk (*) before writing ‘the numbers above…...’. I will just write it with a reduced font size and/or in italics under the table. The asterisk may confuse the audience; thus, they will be looking for asterisk in table. Also see comment for Table 8.

Response: Thank you. We deleted all the asterisk under the tables

  • Table 8 (Line 293):

Please, the single asterisk (*) for the ‘data from a previous study’ is OK. But the double asterisk (**) for CI and Ref is confusing because the double asterisk is not indicated in the table. So, Just write below the table without the double asterisk; CI = confidence interval, Ref = reference.

Response: Thank you. We already changed as you commented.

  • Figure 1 (Line 321-335):

If the authors want to keep the other bps for the ladder label, then this should be properly and nicely done. Please, I do not see the reason why 300bp should be written inside the ladder bands. For me, I am OK with just the 266bp.

Additionally, ‘Sample name was CxT M pool6’ does not read well to me. I will say ‘The positive Culex tritaeniorhynchus sample was obtained from pool6’. CxT M is not part of the figure so there is no need bringing it here; although it is part of Figure 2.

Response: Thank you. We deleted all the bands of the ladder and changed as you commented.

  • Figure 2 (Line 367-412):

I will say ‘The highlight taxa (CxT_M_pool6) in the tree ……’. The authors have forgotten to indicate the members of the respective genotypes (GT I, III, and IV) in the tree. This will make the tree comprehensible by just drawing a line or an arc to indicate this composition. This is very important.

Response: Thank you. Actually, we have done it, but we have no idea why it was disappeared in this version. We already added in the figure.

  • Discussion (Line 416-450):

Between the subtitles ‘mosquito distribution and composition’ and ‘mosquito collection method’, the flow is disrupted. Authors should discuss the ‘mosquito collection method’ before the ‘mosquito distribution and composition’.

Response: Thank you. We already changed the order of the two sections.

  • Line 439:

The statement should read ‘……Cx. quinquefasciatus and other selected species.’

Response: Thank you. We already fixed it.

  • References (Line 589-719):

Although the Authors did a bit of the corrections in the in-text referencing and bibliography. I still have some outstanding issues. Please, I was wondering if the authors used any of the referencing packages such as end-note or mendeley I’ve mentioned earlier. There are still inconsistencies in the bibliography. For example: - ‘doi’ was indicated for some of the journals but some not although they have. For example, Line 712-713 (Schuh et al) has the doi: 10.1128/JVI.02686-13, but this

was not indicated.

- I can see ‘internet’ in sections of references of published articles in recognized journals.

- I can also see ‘Available from’ in the references of published articles in recognized journals which are not supplementary materials, reports, conference proceedings etc. We can use ‘Available from’ for say WHO reports but not for articles in journals.

I know the Authors can do better at the references.

Response: Thank you so much. We used Mendeley but some of articles were the same format. Now, we already fixed all the problems and didn’t use doi for all of references.